# PREFERENCE-BASED MULTI-AGENT REINFORCEMENT LEARNING: DATA COVERAGE AND ALGORITHMIC TECHNIQUES

## ABSTRACT

We initiate the study of Preference-Based Multi-Agent Reinforcement Learning (PbMARL), exploring both theoretical foundations and empirical validations. We define the task as identifying the Nash equilibrium from a preference-only offline dataset in general-sum games, a problem marked by the challenge of sparse feedback signals. Our theory establishes the upper complexity bounds for Nash Equilibrium in effective PbMARL, demonstrating that single-policy coverage is inadequate and highlighting the importance of unilateral dataset coverage. These theoretical insights are verified through comprehensive experiments. To enhance the practical performance, we further introduce two algorithmic techniques. (1) We propose a Mean Squared Error (MSE) regularization along the time axis to achieve a more uniform reward distribution and improve reward learning outcomes. (2) We propose an additional penalty based on the distribution of the data set to incorporate pessimism, improving stability and effectiveness during training. Our findings underscore the multifaceted approach required for PbMARL, paving the way for effective preference-based multi-agent systems.

## 1 INTRODUCTION

Large language models (LLMs) have achieved significant progress in natural language interaction, knowledge acquisition, instruction following, planning and reasoning, which has been recognized as the sparks for AGI (Bubeck et al., 2023). The evolution of LLMs fosters the field of agent systems, wherein LLMs act as the central intelligence (Xi et al., 2023). In these systems, multiple LLMs can interact with each other as well as with external tools. For instance, MetaGPT assigns LLM agents various roles, akin to those in a technology company, enabling them to cooperate on complex software engineering tasks (Hong et al., 2023).

Despite some empirical successes in agent systems utilizing closed-source LLMs, finetuning these systems and aligning them with human preferences remains a challenge. Reinforcement learning from human feedback (RLHF) has played an important role in aligning LLMs with human preferences (Christiano et al., 2017; Ziegler et al., 2019). However, unexpected behavior can arise when multiple LLMs interact with each other. In addition, reward design has been a hard problem in multi-agent reinforcement learning (Devlin et al., 2011). Thus, it is crucial to further align the multi-agent system from preference feedback.

We address this problem through both theoretical analysis and empirical experiments. Theoretically, we characterize the dataset coverage condition for PbMARL that enables learning the Nash equilibrium, which serves as a favorable policy for each player. Empirically, we validate our theoretical insights through comprehensive experiments utilizing the proposed algorithmic techniques.

### 1.1 CONTRIBUTIONS AND TECHNICAL NOVELTIES

**1. Necessary and Sufficient Dataset Coverage Condition for PbMARL.** In single-agent RLHF, (Zhu et al., 2023) demonstrated that single policy coverage is sufficient for learning the optimal policy. However, we prove that this condition no longer holds for PbMARL by providing a counterexample. Instead, we introduce an algorithm that operates under unilateral coverage, a condition derived from

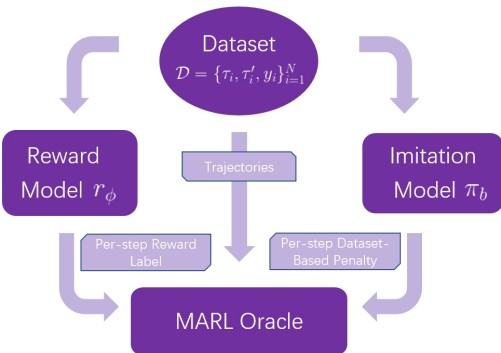

Figure 1: The overall pipeline of offline PbMARL. $\mathcal{D}$ is the preference dataset where $\tau_i, \tau_i'$ are trajectories and $\mathbf{y}_i \in \{1, -1\}^m$ indicates which trajectory is preferred by each agent. $r_\phi$ is the learned reward. $\pi_b$ is the learned reference policy using imitation learning.

offline MARL (Cui and Du, 2022a; Zhong et al., 2022). Specifically, this condition requires the dataset to cover all unilateral deviations from a Nash equilibrium policy. For further details, see Section 4.

**2. Algorithmic Techniques for Practical Performance.** As a foundational exploration into PbMARL research, we focus on employing the simplest learning framework, incorporating only the essential techniques necessary to ensure the approach's feasibility. The framework consists of three key components: 1) leveraging the preference dataset to learn a reward function, 2) mitigating extrapolation errors with pessimism, and 3) determining the final policy. Figure 1 provides an overview of the process.

However, additional algorithmic techniques are required to identify a robust policy, even when the dataset demonstrates good coverage according to our theoretical insights.

- **Reward regularization.** We observed that the reward learned through standard Maximum Likelihood Estimation (MLE) is sparse and spiky, making it difficult for standard RL algorithms to utilize effectively (cf. Figure 2 (b2)). To address this, we introduce an additional Mean Squared Error (MSE) loss between the predictions of adjacent time steps as a form of regularization. This regularization helps to prevent the model from accumulating reward signals solely at the final time step or relying on reward-irrelevant observation patterns, which could otherwise result in the complete failure in producing meaningful predictions.

- **Dataset Distribution-Based Pessimism.** To mitigate the extrapolation error in offline RL, we add an extra reward term based on the density of a certain state-action pair in the dataset to implement pessimism. In our approach, an imitation learning agent is trained to model the density function. The final policy is then trained using a DQN-based Value Decomposition Network (VDN) (Mnih et al., 2013; Sunehag et al., 2017). Our ablation study demonstrates the critical role of appropriately tuning the reward coefficient to ensure training stability and performance (see Table 4).

**3. Experiment Results.** Our experiments, following the pipeline described above, confirm the theoretical necessity of unilateral coverage. We conducted comprehensive ablation studies on three cooperative Multi-Agent Particle Environment (MPE) scenarios (Mordatch and Abbeel, 2017): Spread-v3, Tag-v3, and Reference-v3. These studies focused on the hyperparameter selection for the reward regularization coefficient $\alpha$, pessimism coefficient $\beta$, and dataset diversity. The empirical results (Table 2) demonstrate that: 1) simply adding trivial trajectories to expert demonstrations can enhance performance, 2) unilateral datasets are advantageous, and 3) dataset diversity contributes to lower variance.

Our ablation experiments underscore the effectiveness of the proposed algorithmic techniques. Additionally, we introduced a principled standardization technique that can efficiently tune hyperparameters across all environments and datasets.

## 2 RELATED WORKS

**Reinforcement Learning from Human Feedback (RLHF).** RLHF, or preference-based RL (PbRL), plays a pivotal role in alignment with various tasks such as video games (Warnell et al., 2018; Brown et al., 2019), robotics (Jain et al., 2013; Kupcsik et al., 2016; Christiano et al., 2023; Shin et al., 2023), image augmentation (Metcalf et al., 2024), and large language models (Ziegler et al., 2020; Wu et al., 2021; Nakano et al., 2022; Menick et al., 2022; Stiennon et al., 2022; Bai et al., 2022; Glaese et al., 2022; Ganguli et al., 2022; Ouyang et al., 2022). Additionally, a body of work focuses on the reward models behind preference data (Sadigh et al., 2017; Bıyık and Sadigh, 2018; Gao et al., 2022; Hejna and Sadigh, 2023). Recent works like VIPO (Cen et al., 2024) incorporates uncertainty-aware regularization into the reward model, while (Liu et al., 2024) address over-optimization using adversarial regularization. Direct preference optimization (DPO, Rafailov et al. (2023)) and its variants (Azar et al., 2023; Rafailov et al., 2024) approach RLHF without directly handling the reward model. Theoretical studies have also explored guarantees, such as sample complexity and regret, and the limitations of certain RLHF algorithms (Novoseller et al., 2020; Xu et al., 2020; Pacchiano et al., 2023; Chen et al., 2022; Razin et al., 2023; Zhu et al., 2024a; Wang et al., 2023c; Xiong et al., 2024; Zhu et al., 2024b).

**Offline Reinforcement Learning.** Offline RL (Lange et al., 2012; Levine et al., 2020) has achieved success in a wide range of real-world applications, including robotics (Pinto and Gupta, 2015; Levine et al., 2016; Chebotar et al., 2021; Kumar et al., 2023), healthcare (Raghu et al., 2017; Wang et al., 2018), and autonomous driving (Shi et al., 2021; Lee et al., 2024). Key algorithms such as Behavior Cloning, BRAC (Wu et al., 2019), BEAR (Kumar et al., 2019), and CQL (Kumar et al., 2020; Lyu et al., 2024) have driven these successes. Theoretical research on offline RL has primarily focused on sample complexity under various dataset coverage assumptions Le et al. (2019); Chen and Jiang (2019); Yin et al. (2020); Rashidinejad et al. (2023); Yin et al. (2021; 2022); Shi et al. (2022); Nguyen-Tang et al. (2022); Xie et al. (2022); Xiong et al. (2023b); Li et al. (2024); Xie et al. (2023); Mete et al. (2021).

**Multi-Agent Reinforcement Learning (MARL).** Many real-world scenarios are naturally modeled as multi-agent environments, whether cooperative or competitive. As a result, MARL has gained popularity in video games (Tian et al., 2017; Vinyals et al., 2017; Silver et al., 2017; Vinyals et al., 2019), network design (Shamsoshoara et al., 2018; Kaur and Kumar, 2020), energy sharing (Prasad and Dusparic, 2018), and autonomous driving (Palanisamy, 2019; Yu et al., 2020; Zhou et al., 2022). Prominent algorithms in MARL include IQL (Tan, 2003), MADDPG (Lowe et al., 2020), COMA (Foerster et al., 2017), MAPPO (Yu et al., 2022), VDN (Sunehag et al., 2017), and QMIX (Rashid et al., 2018). Theoretical research has made great process in reducing the sample complexity(Wang et al., 2023b; Xiong et al., 2023a).

**Offline MARL.** Offline MARL is a practical solution for handling sophisticated multi-agent environments. Empirically, to address issues related to out-of-distribution actions and complex reward functions, previous works have developed algorithms such as MABCQ (Jiang and Lu, 2023), ICQ-MA (Yang et al., 2021), OMAR (Pan et al., 2022), and OMIGA (Wang et al., 2023a), which incorporate regularization or constraints on these actions and functions. MOMA-PPO (Barde et al., 2024) is a model-based approach to offline MARL that generates synthetic interaction data from offline datasets. Tseng et al. (2022) combines knowledge distillation with multi-agent decision transformers (Meng et al., 2022) for offline MARL. Theoretical understanding of offline MARL, particularly in the context of Markov games, has been advanced by works that provide sample complexity guarantees for learning equilibria Sidford et al. (2019); Cui and Yang (2020); Zhang et al. (2023a; 2020); Abe and Kaneko (2020); Cui and Du (2022a;b); Zhang et al. (2023b); Blanchet et al. (2023); Shi et al. (2023); Zhong et al. (2022).

# 3 PRELIMINARIES

**General-sum Markov Games.** We consider an episodic time-inhomogeneous general-sum Markov game $\mathcal{M}$, consisting of $m$ players, a shared state space $\mathcal{S}$, an individual action space $\mathcal{A}_i$ for each player $i \in [m]$ and a joint action space $\mathcal{A} = \mathcal{A}_1 \times \mathcal{A}_2 \times \cdots \mathcal{A}_m$. The game has a time horizon $H$, an initial state $s_1$, state transition probabilities $\mathbb{P} = (\mathbb{P}_1, \mathbb{P}_2, \cdots, \mathbb{P}_H)$ with $\mathbb{P}_h : \mathcal{SA} \to \Delta(\mathcal{S})$, and rewards $R = R_h(\cdot \mid s_h, a_h)_{h=1}^H$ where $R_{h,i} \in [0, 1]$ represents the random reward for player $i$ at step $h$. At each step $h \in [H]$, all players observe current state $s_h$ and simultaneously choose their actions $\mathbf{a}_h = (a_{h,1}, a_{h,2}, \cdots, a_{h,m})$. The next state $s_{h+1}$ is then sampled from $\mathbb{P}_h(\cdot \mid s_h, \mathbf{a}_h)$, and the reward $r_{h,i}$ for player $i$ is sampled from $R_{h,i}(\cdot \mid s_h, \mathbf{a}_h)$. The game terminates at step $H + 1$, with each player aiming to maximize the total collected rewards.

We use $\pi = (\pi_1, \pi_2, \cdots, \pi_m)$ to denote a joint policy, where the individual policy for player $i$ is represented as $\pi_i = (\pi_{1,i}, \pi_{2,i}, \cdots, \pi_{H,i})$, with each $\pi_{h,i} : S \to \Delta(A_i)$ defined as the Markov policy for player $i$ at step $h$. The state value function and state-action value function for each player $i \in [m]$ are defined as

$$V_{h,i}^\pi(s_h) := \mathbb{E}_\pi \left[ \sum_{t=h}^H r_{t,i}(s_t, \mathbf{a}_t) \mid s_h \right], \quad Q_{h,i}^\pi(s_h) := \mathbb{E}_\pi \left[ \sum_{t=h}^H r_{t,i}(s_t, \mathbf{a}_t) \mid s_h, \mathbf{a}_h \right],$$

where $\mathbb{E}_\pi = \mathbb{E}_{s_1, \mathbf{a}_1, \mathbf{r}_1, \cdots, s_{H+1} \sim \pi, \mathcal{M}}$ denotes the expectation over the random trajectory generated by policy $\pi$. The best response value for player $i$ is defined as

$$V_{h,i}^{\dagger, \pi_{-i}}(s_h) := \max_{\pi_i} V_{h,i}^{\pi_i, \pi_{-i}}(s_h),$$

which represents the maximal expected total return for player $i$ given that the other players follow policy $\pi_{-i}$.

A Nash equilibrium is a policy configuration where no player has an incentive to change their policy unilaterally. Formally, we measure how closely a policy approximates a Nash equilibrium using the *Nash-Gap*:

$$\text{Nash-Gap}(\pi) := \sum_{i \in [m]} \left[ V_{1,i}^{\dagger, \pi_{-i}}(s_1) - V_{1,i}^\pi(s_1) \right].$$

By definition, the Nash-Gap is always non-negative, and it quantifies the potential benefit each player could gain by unilaterally deviating from the current policy. A policy $\pi$ is considered an $\epsilon$-Nash equilibrium *iff* Nash-Gap$(\pi) \le \epsilon$.

**Offline Multi-agent Reinforcement Learning with Preference Feedback.** In offline MARL with Preference Feedback, the algorithm has access to a pre-collected preference dataset generated by an unknown behavior policy interacting with an underlying Markov game. We consider two sampled trajectories, $\tau = (s_1, \mathbf{a}_1, s_2, \mathbf{a}_2, \cdots, s_{H+1})$ and $\tau' = (s_1', \mathbf{a}_1', s_2', \mathbf{a}_2', \cdots, s_{H+1}')$, drawn from distribution $\mathbb{P}(s_1, \mathbf{a}_1, s_2, \cdots, s_{H+1}) = \Pi_h \pi^b(\mathbf{a}_h \mid s_h) \mathbb{P}(s_{h+1} \mid s_h, \mathbf{a}_h)$ induced by the behavior policy $\pi^b$. In MARLHF, the reward signal is not revealed in the dataset. Instead, each player can observe a binary signal $y_i$ from a Bernoulli distribution following the Bradley-Terry-Luce model (Bradley and Terry, 1952):

$$\mathbb{P}(y_i = 1 \mid \tau, \tau') = \frac{\exp(\sum_{h=1}^H r_i(s_h, \mathbf{a}_h))}{\exp(\sum_{h=1}^H r_i(s_h, \mathbf{a}_h)) + \exp(\sum_{h=1}^H r_i(s_h', \mathbf{a}_h'))}, \forall i \in [m].$$

We make the standard linear Markov game assumption (Zhong et al., 2022):

**Assumption 1.** $\mathcal{M}$ *is a linear Markov game with a feature map* $\psi : \mathcal{S} \times \mathcal{A} \to \mathbb{R}^d$ *if we have*

$$\mathbb{P}_h(s_{h+1} \mid s_h, \mathbf{a}_h) = \langle \psi(s_h, \mathbf{a}_h), \mu_h(s_{h+1}) \rangle, \forall (s_h, \mathbf{a}_h, s_{h+1}, h) \in \mathcal{S} \times \mathcal{A} \times \mathcal{S} \times [H],$$

$$r_i(s_h, \mathbf{a}_h) = \langle \psi(s_h), \theta_{h,i} \rangle, \forall (s_h, \mathbf{a}_h, h, i) \in \mathcal{S} \times \mathcal{A} \times [H] \times [m],$$

*where* $\mu_h$ *and* $\theta_{h,i}$ *are unknown parameters. Without loss of generality, we assume* $\|\psi(s, \mathbf{a})\| \le 1$ *for all* $(s, \mathbf{a}) \in \mathcal{S} \times \mathcal{A}$ *and* $\|\mu_h(s)\| \le \sqrt{d}, \|\theta\|_h \le \sqrt{d}$ *for all* $h \in [H]$.

The one-hot feature map is defined as $\overline{\psi}_h(s, \mathbf{a}) := [0, \cdots, 0, \psi(s, \mathbf{a}), 0, \cdots, 0] \in \mathbb{R}^{Hd}$, where $\psi(s, \mathbf{a})$ is at position $(h-1)d + 1$ to $hd$.

**Value-Decomposition Network (VDN).** In our experiments, we utilize VDN as an offline MARL algorithm for its effectiveness and simplicity. VDN (Sunehag et al., 2017) is a Q-learning style MARL architecture for cooperative games. It takes the idea of decomposing the team value function into agent-wise value functions, expressed as: $Q_h(s, \mathbf{a}) = \sum_{i=1}^{n} Q_{h,i}(s, a_i)$. In our experiments, we applied Deep Q-Network (DQN) (Mnih et al., 2013) with VDN to learn the team Q function. We chose DQN to maintain the simplicity and controllability of the experimental pipeline, which facilitates a more accurate investigation of the impact of various techniques on the learning process.

## 4 DATASET COVERAGE THEORY FOR MARLHF

In this section, we study the dataset coverage assumptions for offline MARLHF. For offline single-agent RLHF, Zhu et al. (2023); Zhan et al. (2023) show that single policy coverage is sufficient for learning the optimal policy. However, we prove that this assumption is insufficient in the multi-agent setting by constructing an counterexample. In addition, we prove that unilateral policy coverage is adequate for learning the Nash equilibrium.

### 4.1 POLICY COVERAGES

We quantify the information contained in the dataset using covariance matrices, as the rewards and transition kernels are parameterized by a linear model. With a slight abuse of the notation, for trajectory $\tau = (s_1, \mathbf{a}_1, s_2, \mathbf{a}_2, \cdots, s_{H+1})$, we use $\psi(\tau) := [\psi(s_1, \mathbf{a}_1), \psi(s_2, \mathbf{a}_2), \cdots, \psi(s_H, \mathbf{a}_H)]$ to denote the concatenated trajectory feature. The reward coverage is measured by the preference covariance matrix:

$$\Sigma_{\mathcal{D}}^r = \lambda I + \sum_{(\tau, \tau') \in \mathcal{D}} (\psi(\tau) - \psi(\tau'))(\psi(\tau) - \psi(\tau'))^\top,$$

where $\psi(\tau) - \psi(\tau')$ is derived from the preference model. Similarly, the transition coverage is measured by the covariance matrix:

$$\Sigma_{\mathcal{D},h}^{\mathbb{P}} = \lambda I + \sum_{(\tau, \tau') \in \mathcal{D}} \left[ \psi(s_h, \mathbf{a}_h)\psi(s_h, \mathbf{a}_h)^\top + \psi(s_h', \mathbf{a}_h')\psi(s_h', \mathbf{a}_h')^\top \right].$$

For a given state and action pair $(s_h, \mathbf{a}_h)$, the term $\left\| \overline{\psi}_h(s_h, \mathbf{a}_h) \right\|_{[\Sigma_{\mathcal{D}}^r]^{-1}}$ measures the uncertainty in reward estimation and $\| \psi(s_h, \mathbf{a}_h) \|_{[\Sigma_{\mathcal{D},h}^{\mathbb{P}}]^{-1}}$ measures the uncertainty in transition estimation. As a result, the overall uncertainty of a given policy $\pi$ with dataset $\mathcal{D}$ is measured by

$$U_{\mathcal{D}}(\pi) := \mathbb{E}_\pi \left[ \sum_{h=1}^{H} \left\| \overline{\psi}_h(s_h, a_h) \right\|_{[\Sigma_{\mathcal{D}}^r]^{-1}} + \sum_{h=1}^{H} \| \psi(s_h, a_h) \|_{[\Sigma_{\mathcal{D},h}^{\mathbb{P}}]^{-1}} \right].$$

**Definition 1.** *For a Nash equilibrium $\pi^*$, different policy coverages are measured by the following quantities:*

- *Single policy coverage: $U_{\mathcal{D}}(\pi^*)$.*

- *Unilateral policy coverage: $\max_{i, \pi_i} U_{\mathcal{D}}(\pi_i, \pi_{-i}^*)$.*

- *Uniform policy coverage: $\max_\pi U_{\mathcal{D}}(\pi)$.*

*Intuitively, small $U_{\mathcal{D}}(\pi^*)$ indicates that the dataset contains adequate information about $\pi^*$. A small $\max_{i, \pi_i} U_{\mathcal{D}}(\pi_i, \pi_{-i}^*)$ implies that the dataset covers all of the unilateral deviations of $\pi^*$, and small $\max_\pi U_{\mathcal{D}}(\pi^*)$ suggests that the dataset covers all possible policies.*

### 4.2 SINGLE POLICY COVERAGE IS INSUFFICIENT

Our objective is to learn a Nash equilibrium policy from the dataset, which necessitates that the dataset sufficiently covers the Nash equilibrium. In the single-agent scenario, if the dataset covers the

optimal policy, pessimism-based algorithms can be employed to recover the optimal policy. However, previous work (Cui and Du, 2022a; Zhong et al., 2022) has demonstrated that single policy coverage is insufficient for offline MARL. We extend this result to the context of offline MARL with preference feedback, as follows:

**Theorem 1.** *(Informal) If the dataset only has coverage on the Nash equilibrium policy (i.e. small $U_\mathcal{D}(\pi^*)$), it is not sufficient for learning an approximate Nash equilibrium policy.*

The proof is derived by a reduction from standard offline MARL to MARLHF. Suppose that MARLHF with single policy coverage suffices, we could construct an algorithm for standard offline MARL, which leads to a contradiction. The formal statement and the detailed proof are deferred to Appendix A.1.

### 4.3 UNILATERAL POLICY COVERAGE IS SUFFICIENT

While single policy coverage is too weak to learn a Nash equilibrium, uniform policy coverage, though sufficient, is often too strong and impractical for many scenarios. Instead, we focus on unilateral policy coverage, which offers a middle ground between single policy coverage and uniform policy coverage.

**Theorem 2.** *(Informal) If the dataset has unilateral coverage on the Nash equilibrium policy, there exists an algorithm that can output an approximate Nash equilibrium policy.*

The detailed proof is deferred to Appendix A.2. We leverage a variant of Strategy-wise Bonus and Surrogate Minimization (SBSM) algorithm in (Cui and Du, 2022b) with modified policy evaluation and policy optimization subroutines. Intuitively, the algorithm identifies a policy that minimizes a pessimistic estimate of the Nash gap. As a result, if the dataset has unilateral coverage, the output policywill have a small Nash gap and serves as a good approximation of the Nash equilibrium.

## 5 ALGORITHMIC TECHNIQUES FOR PRACTICAL PERFORMANCE

In Section 4, we provided a theoretical characterization of the dataset requirements for MARLHF. However, the algorithm used in Theorem 2 is not computationally efficient. In this section, we propose a practical algorithm for MARLHF and validate our theoretical findings through experiments.

### 5.1 HIGH-LEVEL METHODOLOGY

Our MARLHF pipeline consists of two phases: In the first step, we train a reward prediction model $\phi$ and approximate the behavior policy $\pi_b$ using imitation learning; in the second step, we then apply an MARL algorithm to maximize a combination of the KL-divergence-based reward and standardized predicted reward $r_\phi$, ultimately deriving the final policy $\pi_\mathbf{w}$.

**Step 1: Reward Training and Dataset Modeling.** Given the preference signals of trajectories, we use neural networks to predict step-wise rewards $r_\phi(s_h, a_h)$ for each agent, minimizing the loss defined in (1). The objective is to map $(s, a_i)$-pairs to reward values such that the team returns align with the preference signals. At the same time, in order to utilize distribution-based penalty term $\log \pi_b(s, a)$ to cope with the extrapolation error in offline learning, an imitation learner is trained over the entire dataset to model the behavior policy $\pi_b$.

**Step 2: Offline MARL.** Although in this work, VDN is chosen as the MARL oracle, it should be noted that other MARL architectures are also applicable. With the reward model $r_\phi$ and the approximated dataset distribution learned in Step 1, we are now able to construct a virtual step-wise reward for each agent. The agents are then trained to maximize the target defined in (3).

Given this framework, additional techniques are required to build a strong practical algorithm, which we provide more details below.

## 5.2 REWARD REGULARIZATION

Compared to step-wise reward signals, preference signals are $H$ times sparser, making them more challenging for a standard RL algorithm to utilize effectively. Concretely, this reward sparsity causes the naive optimization of the negative log-likelihood (NLL) loss to suffer from two key problems:

1. **Sparse and spiky reward output.** When calculating NLL losses, spreading the reward signal along the trajectories is equivalent to summing it at the last time step (Figure 2a). However, a sparse reward signal is harder for traditional RL methods to handdle due to the lack of continuous supervision. More uniformly distributed rewards across the entire trajectory generally leads to more efficient learning in standard RL algorithms.

2. **Over-reliance on irrelevant features.** The model may exploit redundant features as shortcuts to predict rewards. For instance, expert agents in cooperative games usually exhibit a fixed pattern of collaboration from the very beginning of the trajectory (such as specific actions or communication moves). The reward model might use these patterns to differentiate them from agents of other skill levels, thereby failing to capture the true reward-observation causal relationships.

To mitigate these problems, we introduce an extra Mean Squared Error (MSE) regularization along the time axis (Equation 1, 2). By limiting the sudden changes in reward predictions between adjacent time steps, this regularization discourages the reward model from concentrating its predictions on just a few time steps. While these issues can also be mitigated by using more diversified datasets and adding regularization to experts to eliminate reward-irrelevant action patterns, these approaches can be costly and sometimes impractical in real-world applications. In contrast, our MSE regularization is both easy to implement and has been empirically verified to be effective, creating more uniform reward distribution (Figure 2) and better performances.

$$L_{\text{RM}}(\phi) = -\mathbb{E}_{\mathcal{D}} \left[ \sum_{i=1}^{m} \log \sigma(y_i(r_{\phi,i}(\tau_1) - r_{\phi,i}(\tau_2))) \right] + \frac{\alpha}{\text{Var}_{\mathcal{D}}(r_\phi)} L_{\text{MSE}}(\phi, \tau), \quad (1)$$

where the regularization term $L_{\text{MSE}}$ is defined as:

$$L_{\text{MSE}}(\phi, \tau) = \mathbb{E}_{\mathcal{D}} \left[ \sum_{h=1}^{H-1} \|r_\phi(s_h, \mathbf{a}_h) - r_\phi(s_{h+1}, \mathbf{a}_{h+1})\|_2^2 \right]. \quad (2)$$

Here $\alpha$ is the regularization coefficient, which is set to be 1 in our experiments. The variance of $r_\phi$ is calculated over the training set to adaptively scale the regularization term. During training, $\text{Var}_{\mathcal{D}}(r_\phi)$ is detached to prevent gradients from flowing through it. The effectiveness of this method is validated in the ablation study (cf. Section 6.3).

## 5.3 DATASET DISTRIBUTION-BASED PESSIMISM

There are various methods to mitigate the over-extrapolation errors in offline RL (Peng et al., 2019; Nair et al., 2021), including conservative loss over the Q-function (Kumar et al., 2020) and directly restricting the learned policy actions to those within the dataset (Fujimoto et al., 2019). We add a per-step dataset-based penalty term, $\log \pi_b(s, \mathbf{a})$, as pessimism towards less explored states. Imitation learning is utilized to estimate the behavior policy $\pi_b$ from the dataset distribution. To stabilize training, we standardize predicted reward $r_\phi$ over $\mathcal{D}$ before combining it with the penalty term to make them comparable:

$$\text{objective}(\mathbf{w}) = \mathbb{E}_{\tau \sim \pi_{\mathbf{w}}} \left[ \sum_{h=1}^{H} r_{\text{std}}(s_h, \mathbf{a}_h, \phi) + \text{clip}(\beta \log \pi_b(s_h, \mathbf{a}_h), -10, 1) \right], \quad (3)$$

where $\beta$ is the pessimism coefficient, set to be $(1, 1, 10, 10)$ in Spread-v3, Reference-3, Tag-v3 and Overcooked respectively in the main experiments. The clip operator is defined by $\text{clip}(x, a, b) = \min(b, \max(a, x))$. The standardized reward $r_{\text{std}}$ is defined as:

$$r_{\text{std}}(s_h, \mathbf{a}_h, \phi) = \sum_{i=1}^{m} \frac{r_\phi(s_h, a_{h,i}) - \mathbb{E}_{\mathcal{D}}(r_\phi)}{\sqrt{\text{Var}_{\mathcal{D}}(r_\phi)}}. \quad (4)$$

Intuitively, the penalty term $\log \pi_b(s_h, \mathbf{a}_h)$ discourages the agents from deviating from the most preferred actions in the dataset. The effectiveness of this method is validated in the ablation study (cf. Section 6.4).

# 6 EXPERIMENTS

We design a series of experiments to validate our theories and methods in common general-sum games. Specifically, we first use online RL algorithms to train expert agents, and take intermediate checkpoints as rookie agents. Then, we use these agents to collect datasets and use the Bradley-Terry model over standardized returns to simulate human preference. Experiments are carried out to verify the efficiency of our approach with unilateral policy dataset coverage (in Theorem 2) while single policy coverage is insufficient (stated in Theorem 1). We also design ablation studies to showcase the importance of our methods, particularly focusing on reward regularization and dataset distribution-based pessimism.

## 6.1 ENVIRONMENTS

Our experiments involved 3 Multi-Agent Particle Environments (MPE), including Spread-v3, Tag-v3 and Reference-v3, and Overcooked environment implemented with JaxMARL codebase (Rutherford et al., 2023). **Spread-v3** contains a group of agents and target landmarks, where the objective is to cover as many landmarks as possible while avoiding collisions. **Tag-v3** contains two opposing groups, where quicker "preys" need to escape from "predators". To ensure a fair comparison of different predator cooperation policies, we fixed a pretrained prey agent. **Reference-v3** involves two agents and three potential landmarks, where the agents need to find each one's target landmark to receive a high reward. The target landmark of each agent is only known by the other agent at first. **Overcooked** involves two agents moving and operating objects in a gridworld. A more detailed description of the tasks and their associated challenges is provided in Appendix B.2.

## 6.2 THE IMPORTANCE OF DATASET DIVERSITY

To study the influence of diversity of dataset, we manually designed 4 kinds of mixed joint behavior polices, and change their ratios to form different datasets.

- Expert policy: $n$ expert agents. Trained with online RL algorithms till convergence.
- Rookie policy: $n$ rookie agents. Trained with online RL algorithms with early stop.
- Trivial policy: $n$ random agents. All actions are uniformly sampled from the action space.
- Unilateral policy: $n - 1$ expert agents and 1 rookie agent of different proficiency level.

Table 1 presents the ratio of trajectories collected by the four different policies. The experiments are designed to hierarchically examine the roles of diversity (Diversified vs. Mix-Unilateral), unilateral coverage (Mix-Unilateral vs. Mix-Expert), and trivial comparison (Mix-Expert vs. Pure-Expert).

The ranking of diversity follows the order:

$$\text{Pure-Expert} < \text{Mix-Expert} < \text{Mix-Unilateral} < \text{Diversified}$$

Due to the inherent limitations of offline reinforcement learning (RL) in action selection dictated by the dataset, the effectiveness of learning is often strongly correlated with dataset quality, i.e. the level of expertise demonstrated in the dataset. However, the results in preference-based MARL experiments partially diverge from this conventional conclusion. While the quality of the dataset remains critical, experiments on Reference-v3 and Overcooked (Table 2) indicate that diversity and unilateral data can significantly enhance the performance of the reward model, thereby facilitating learning.

The main experimental results are presented in Table 2 and Table 3. Among all the experiments, apart from the experiments on Tag-v3, where the high operational precision requirements make data quality more critical than diversity, the other three environments validate our conclusions across all algorithms.

|  | Expert | Unilateral | Rookie | Trivial |
|---|---|---|---|---|
| Diversified | 1 | 1 | 1 | 1 |
| Mix-Unilateral | 2 | 1 | 0 | 1 |
| Mix-Expert | 3 | 0 | 0 | 1 |
| Pure-Expert | 4 | 0 | 0 | 0 |

Table 1: Final datasets mixed with various ratios. The overall dataset size is kept to 38400 trajectories for MPE, and 960 trajectories for Overcooked. (cf. B.1)

| Algorithm | Dataset | Spread-v3 | Tag-v3 | Reference-v3 | Overcooked |
|---|---|---|---|---|---|
| VDN with Pessimism Penalty | Diversified | -21.16 $_{\pm 0.54}$ | 29.28 $_{\pm 1.08}$ | -18.89 $_{\pm 0.60}$ | **238.89** $_{\pm \textbf{3.50}}$ |
|  | Mix-Unilateral | -21.03 $_{\pm 0.44}$ | 36.65 $_{\pm 0.70}$ | -18.80 $_{\pm 0.63}$ | 221.80 $_{\pm 26.66}$ |
|  | Mix-Expert | -20.98 $_{\pm 0.54}$ | 35.96 $_{\pm 0.86}$ | -18.80 $_{\pm 0.44}$ | 35.26 $_{\pm 55.19}$ |
|  | Pure-Expert | -21.01 $_{\pm 0.57}$ | **39.55** $_{\pm \textbf{0.77}}$ | -28.97 $_{\pm 2.89}$ | 3.36 $_{\pm 7.19}$ |

Table 2: In the simplest environment, Spread-v3, different dataset gives similar performance. In Tag-v3 environment, where precise actions are required, the quality of the dataset (proportion of expert demonstration) is more important than diversity. In contrast, in Overcooked environment, which focuses on strategy learning and demands less on precision, dataset diversity contributes to improved stability, with Unilateral playing a particularly critical role. In the Reference-v3 environment, which balances the need for precision and strategic, the importance of both factors is more balanced, but non-expert data is still necessary.

### 6.3 EXPERIMENTS FOR REWARD REGULARIZATION

In Figure 2, we examined the effectiveness of our proposed reward regularization technique. Figure 2a demonstrates that without regularization, the learned rewards tend to be sparse and spiky compared to the ground truth rewards.

We also observe that the rewards often exhibit temporal continuity, which can create greater discrepancies with the sparse, pulse-like ground truth. Notably, we found that adding stronger regularization does not necessarily lead to underfitting of the reward model; in some cases, it even helps the model converge to a lower training loss. Detailed parameters and experimental results are provided in the appendix (cf. Table 8). We attribute this to the role of regularization in preventing the model from overly relying on shortcuts.

### 6.4 OTHER ABLATION STUDIES

**Pessimism coefficient**    Due to the clipping in 3, excessively large $\beta$ values will not dominate the entire reward function. As a result, larger $\beta$ values almost never degrade the agent's performance in our experiments (Table 4). This allows us to increase $\beta$ with relative confidence. Therefore, we generally recommend setting $\beta$ to a value between 10 and 100 for optimal performances.

**Scalability**    We also tested the scalability on Spread-v3. While our current approach manages the scaling of agents without introducing new problems, it does not specifically address the inherent issues of instability and complexity that are well-documented in traditional MARL (cf. Appendix B.4).

## 7 DISCUSSION

In this paper, we proposed dedicated algorithmic techniques for offline PbMARL and provided theoretical justification for the unilateral dataset coverage condition. We believe our work is a significant step towards systematically studying PbMARL and offers a foundational framework for future research in this area. The flexibility of our framework allows for application across a wide

| Algorithm | Dataset | Spread-v3 | Reference-v3 | Overcooked |
|---|---|---|---|---|
| MAIQL | Diversified | -25.33 $_{\pm 1.40}$ | -22.15 $_{\pm 0.55}$ | **16.59** $_{\pm 11.22}$ |
| | Mix-Unilateral | -23.25 $_{\pm 1.06}$ | -23.22 $_{\pm 1.37}$ | 0.00 $_{\pm 0.00}$ |
| | Mix-Expert | -23.26 $_{\pm 0.90}$ | -24.21 $_{\pm 1.60}$ | 0.00 $_{\pm 0.00}$ |
| | Pure-Expert | -26.01 $_{\pm 1.53}$ | -29.47 $_{\pm 1.65}$ | 0.00 $_{\pm 0.00}$ |
| MABCQ | Diversified | -20.02 $_{\pm 0.64}$ | -17.64 $_{\pm 0.43}$ | **239.34** $_{\pm 1.67}$ |
| | Mix-Unilateral | -19.47 $_{\pm 0.33}$ | -17.64 $_{\pm 1.11}$ | 215.01 $_{\pm 65.43}$ |
| | Mix-Expert | -19.42 $_{\pm 0.17}$ | -17.88 $_{\pm 0.78}$ | 50.32 $_{\pm 82.82}$ |
| | Pure-Expert | -20.56 $_{\pm 0.38}$ | -25.90 $_{\pm 1.11}$ | 1.14 $_{\pm 3.46}$ |

Table 3: Test returns of MAIQL and MABCQ. In the experimental results, we can observe a clear preference toward more diversified datasets. Compared to our method and BCQ, which directly calculate $\max_a Q$ for Bellman updates, IQL employs expectile regression to estimate it. So MAIQL demands higher accuracy of the reward model. Consequently, the performance improvements brought by dataset diversity are also more pronounced in MAIQL experiments.

| | $\beta = 0$ | $\beta = 0.1$ | $\beta = 1$ | $\beta = 10$ | $\beta = 100$ | $\alpha = 0$ |
|---|---|---|---|---|---|---|
| Spread-v3 | -22.56 $_{\pm 1.61}$ | -22.03 $_{\pm 0.67}$ | -20.82 $_{\pm 0.53}$ | -20.46 $_{\pm 0.51}$ | -20.35 $_{\pm 0.43}$ | -22.21 $_{\pm 0.72}$ |
| Tag-v3 | 4.11 $_{\pm 1.66}$ | 4.25 $_{\pm 0.53}$ | 10.96 $_{\pm 1.20}$ | 28.88 $_{\pm 1.02}$ | 29.53 $_{\pm 1.35}$ | 30.77 $_{\pm 0.57}$ |
| Reference-v3 | -19.69 $_{\pm 0.36}$ | -19.37 $_{\pm 0.53}$ | -18.89 $_{\pm 0.78}$ | -18.33 $_{\pm 0.42}$ | -18.54 $_{\pm 0.46}$ | -21.86 $_{\pm 0.73}$ |
| Overcooked | 0.00 $_{\pm 0.00}$ | 0.00 $_{\pm 0.00}$ | 149.53 $_{\pm 86.74}$ | 238.89 $_{\pm 3.50}$ | **240** $_{\pm 0.00}$ | **240** $_{\pm 0.00}$ |

Table 4: Comparison of test return with different hyperparameters. Standard pipeline take pessimism coefficient $\beta = 1$ for Spread-v3, Reference-v3 and $\beta = 10$ for Tag-v3, Overcooked, and the MSE reward regularization coefficient $\alpha$ is set to the optimal value for fixed $\beta$. All the agents are trained on Diversified Dataset across 10 random seeds. Results show that larger $\beta$ always gives better performance and a proper positive $\alpha$ can improve performance.

range of general games, and our empirical results validate the effectiveness of our proposed methods in various scenarios.

Looking ahead, there is significant potential to extend this work to more complex, real-world scenarios, particularly by integrating Large Language Models (LLMs) into multi-agent systems. Future research will focus on fine-tuning and aligning LLMs within PbMARL, addressing challenges such as increased complexity and the design of effective reward structures.

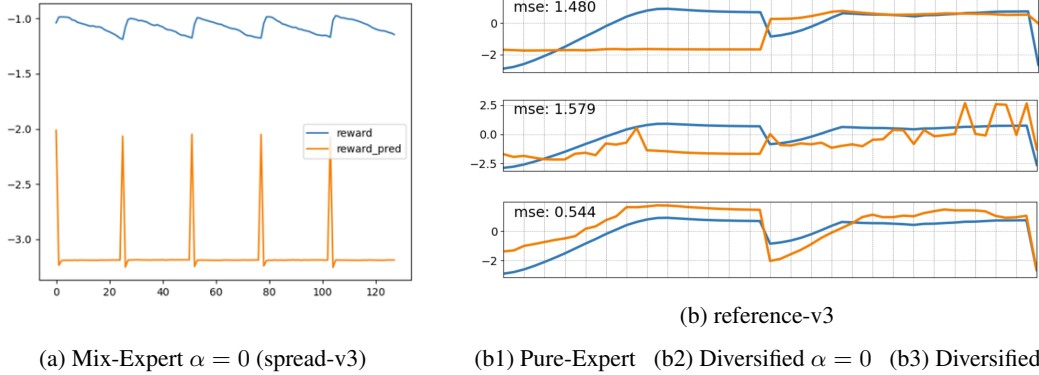

(a) Mix-Expert $\alpha = 0$ (spread-v3)

(b) reference-v3

(b1) Pure-Expert  (b2) Diversified $\alpha = 0$  (b3) Diversified

Figure 2: (a) Averaged reward predictions and ground truth of a trajectory sample on spread-v3. (b) Standardized reward predictions and ground truth of a trajectory sample in reference-v3. When trained with expert data only (b1), $\phi$ experiences a mode collapse, failing to give informative signals. Reward function trained without regularization (b2) shows spiky patterns and tends to accumulate predictions at certain time steps when trained with less diversified datasets as (a). Our method with diversified dataset (b3) gives predictions that approximate the ground truth well.

# 8 REPRODUCIBILITY STATEMENT

All code used for our experiments is included in the supplementary material (`codebase.zip`). Appendix A provides detailed proofs of the theoretical bounds, along with necessary assumptions. Key experimental details and hyperparameters are also outlined in Appendix B. We believe these resources provide a comprehensive foundation for reproducing both the theoretical and empirical results presented in this work.

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
