# OpenReview forum: "Multi-Agent Reinforcement Learning from Human Feedback: Data Coverage and Algorithmic Techniques"
_ICLR.cc/2025/Conference — Submitted to ICLR 2025_

### Official Review · Reviewer_TKUr · 2024-10-28

**Soundness:** 2
**Presentation:** 2
**Contribution:** 3
**Rating:** 5
**Confidence:** 3

**Summary:**

The paper seeks to establish theoretical foundations and make empirical validations for the new research field, Multi-Agent Reinforcement Learning from Human Feedback (MARLHF). The core theoretical contribution is proving that single-policy coverage is insufficient for learning approximate Nash equilibrium policies and that unilateral policy coverage is sufficient to do so. The empirical contribution lies in two techniques, namely, reward regularization which smoothens the reward distribution, and dataset distribution-based pessimism which handles the extrapolation errors. The experiments are designed to verify the correctness of the theoretical claims and the effectiveness of the empirical techniques.

**Strengths:**

- I am not an expert in RLHF, but to my best knowledge, this is the first work for aligning multi-agent systems with human feedback.
- The theoretical claims are concise and seems to be practically useful.
- The experiments are well designed for the purpose of verifying the proposed theoretical claims and empirical techniques.

**Weaknesses:**

The experiments are conducted on a limited range of tasks, which may not be sufficient to verify the generality of the theoretical claims and empirical techniques.

As far as I can tell, there are no other obvious weaknesses of this paper. Potential weaknesses concerning the consistency between the experiment results and the corresponding conclusions are listed as questions below.

**Questions:**

- Figure 1: $\pi_{ref}$, while mentioned in the caption, doesn't seem to be appearing in the figure. Do you mean $\pi_b$?
- What does the blue text mean in Lines 300-301?
- Table 2: The claim in the capture, namely, "in more challenging environments, such as Tag-v3, dataset diversity plays a substantially more significant role", seems inconsistent with the data in the table, where both the mean and the variance of the return of Tag-v3 reach their best in the Pure-Expert dataset which has the least diversity.
- Table 2: The claim in Lines 419-420, namely, "In more challenging tasks, as reflected by higher MSE, the importance of unilateral coverage and diversity becomes more pronounced.", does not seem very obvious from the table, where the diversified and the mix-unilateral dataset achieve the best performance when (Spread-v3 for Mix-unilateral and Reference-v3 for Diversified) the corresponding MSE is low.
- Table 3: Why does setting $\beta$ to a magnitude as large as 100 yield such good results? Doesn't the penalty term completely dominate the loss? Further, it seems strange to me that setting $\beta$ across such a wide range (from 1 to 100) can yield almost the same result, especially when the dataset is the diversified one which contains a large fraction of low-return trajectories.
- Figure 2: What does the x-axis represent?

---

> ### Author Response · Authors · 2024-11-25
>
> #### W1. The experiments are conducted on a limited range of tasks.
> Thank you for the advice! In the updated paper, experiments on Overcooked are added. For other algorithms, experiments with MAIQL and MABCQ are included. These two algorithms are commonly used in offline MARL papers. See results below and more details in Section 6 of the updated paper.
>
> #### Q3 & Q4. Inconsistency between Table 2 and claims.
> We have corrected the corresponding statement in the updated version of our paper.
> The current data is accurate, but the claim "in more challenging environments..." was based on data from a previous version. We updated the experimental data before submission but mistakenly submitted an outdated text version.
>
> According to the earlier data, the expert in the tag environment was relatively weaker, and a more diverse dataset indeed resulted in higher returns. However, for fairness, we switched to a stronger expert, which significantly improved the results of imitation learning (high-$\beta$ settings) while reducing the impact of diversity on the improvement of the reward model.
>
> #### Q5. Table 3: Why does setting to a magnitude as large as 100 yield such good results?
> Thank you for pointing this out. The original paper missed the explanation. In our actual experiments, we clipped $\beta \log \pi(s,a)$ to $[-10, 1]$, which was not mentioned. The clipping serves to smooth out the differences in penalty terms among behaviors with relatively high density in the dataset. After clipping, excessively high $\beta$ values no longer dominate the loss, and the performance for higher $\beta$ becomes very similar. We have added this clarification in the updated version of the paper.
>
> #### Q6. Figure 2: What does the x-axis represent?
> The X-axis in Figure 2 represents time, so all the graphs show the reward/predicted reward curves over time. Their integral corresponds to the return.
>
> #### Typos in the paper
> Thank you for pointing out the typos in our paper, we have corrected them now:
> - Figure 1: Replace $\pi_{ref}$ with $\pi_b$.
> - Line 300: Delete the blue words.

---

> > ### Author Response · Authors · 2024-11-25
> >
> > | Algorithm                | Dataset          | Spread-v3        | Reference-v3     | Overcooked        |
> > |--------------------------|------------------|------------------|------------------|-------------------|
> > | **VDN with Pessimism Penalty** | Diversified      | -21.16 ± 0.54    | -18.89 ± 0.60    | **238.89 ± 3.50** |
> > |                          | Mix-Unilateral   | -21.03 ± 0.44    | -18.80 ± 0.63    | 221.80 ± 26.66    |
> > |                          | Mix-Expert       | -20.98 ± 0.54    | -18.80 ± 0.44    | 35.26 ± 55.19     |
> > |                          | Pure-Expert      | -21.01 ± 0.57    | -28.97 ± 2.89    | 3.36 ± 7.19       |
> > | **MAIQL**                | Diversified      | -25.33 ± 1.40    | -22.15 ± 0.55    | **16.59 ± 11.22** |
> > |                          | Mix-Unilateral   | -23.25 ± 1.06    | -23.22 ± 1.37    | 0.00 ± 0.00       |
> > |                          | Mix-Expert       | -23.26 ± 0.90    | -24.21 ± 1.60    | 0.00 ± 0.00       |
> > |                          | Pure-Expert      | -26.01 ± 1.53    | -29.47 ± 1.65    | 0.00 ± 0.00       |
> > | **MABCQ**                | Diversified      | -20.02 ± 0.64    | -17.64 ± 0.43    | **239.34 ± 1.67** |
> > |                          | Mix-Unilateral   | -19.47 ± 0.33    | -17.64 ± 1.11    | 215.01 ± 65.43    |
> > |                          | Mix-Expert       | -19.42 ± 0.17    | -17.88 ± 0.78    | 50.32 ± 82.82     |
> > |                          | Pure-Expert      | -20.56 ± 0.38    | -25.90 ± 1.11    | 1.14 ± 3.46       |

---

> > > ### Comment · Reviewer_TKUr · 2024-11-25
> > >
> > > Thank you for the clarification and additional results. Most of my concerns are addressed. I've raised my confidence to 3 but decided to keep the score.
> > >
> > > **About the additional task Overcook**
> > >
> > > Seems like finding a proper task where the empirical result aligns with the theoretical claims is not a trivial task itself.... The explanation that this task "focuses on strategy learning and demands less on precision" is somewhat also ambigious to me. Maybe there are more tasks in JaxMARL that can demonstrate the importance of unilateral coverage? Hope to see more convicing task performances in your future versions.
> > >
> > > **About the MSE results**
> > >
> > > Would you mind explaining why the MSE results in Table 2 are deleted in the updated paper?

---

> > > > ### Author Response · Authors · 2024-11-26
> > > >
> > > > Thank you for your positive feedback!
> > > >
> > > > #### Additional task
> > > >
> > > > Specifically, the three MPE environments occur in a continuous state space and focus on location control, where agents need to learn non-trivial force control. On the other hand, Overcooked occurs in a discrete state space, emphasizing strategy learning, with a more direct relationship between actions and dynamics.
> > > >
> > > > Indeed, selecting an environment that is both simple yet non-trivial is not an easy task. The setup of PbMARL poses significant disadvantages in environments with long episodes and complex reward structures, which can dominate the experimental results and make it difficult to reflect differences brought by the datasets. We will continue exploring suitable environments for further experiments.
> > > >
> > > > ##### MSE results
> > > > Considering the page limitations, we have removed the related results and discussions from the main text.
> > > >
> > > > Another consideration is that, since different reward assigning methods can lead to similar optimal policies and trajectory preferences, the MSE between predicted and actual rewards may not always effectively reflect the quality of the reward model in PbRL. For example, in the Overcooked environment, assigning a reward to **cooking the dish** and assigning a reward to **serving the dish** results in very similar returns, as a complete scoring period involves both operations. However, these two reward functions will have a squared difference of 2.
> > > > To avoid potential misunderstandings, we decided to exclude it from the main text.
> > > >
> > > > You can still find the relevant discussions and experimental results in Appendix B.5.
> > > >
> > > > |                | Spread-v3 | Tag-v3 | Reference-v3 | Overcooked |
> > > > |----------------|-----------|--------|--------------|------------|
> > > > | Diversified    | 0.434     | 1.46   | 1.19         | 2.04       |
> > > > | Mix-Unilateral | 0.647     | 1.52   | 1.09         | 1.98       |
> > > > | Mix-Expert     | 0.578     | 1.78   | 1.09         | 2.17       |
> > > > | Pure-Expert    | 0.673     | 1.48   | 2.33         | 1.72       |

---

> > > > > ### Comment · Reviewer_TKUr · 2024-11-27
> > > > >
> > > > > Thank you for the insightful explanation on task selection. The example of dish cooking/serving is intuitive and makes sense to me. Indeed, you can never be too careful when establishing causal relationships.

---

> > > > > > ### Author Response · Authors · 2024-12-04
> > > > > >
> > > > > > Thank you for your thoughtful response. We are pleased to hear that the new examples helped clarify things. We greatly appreciate the reviewer for pointing these out, as it has been instrumental in improving our paper.

---

### Official Review · Reviewer_oCgR · 2024-10-30

**Soundness:** 3
**Presentation:** 3
**Contribution:** 2
**Rating:** 5
**Confidence:** 3

**Summary:**

This study introduces Multi-Agent Reinforcement Learning from Human Feedback (MARLHF) to find Nash equilibria from preference-based data with sparse feedback. A key technique in this paper is to use the MSE regularization for uniform rewards and a pessimism-based penalty—to improve stability and performance, enabling more effective preference-based multi-agent systems.

**Strengths:**

The theoretical analysis presented in this paper is solid and clear, providing a sound theoretical bound for the proposed method to solve MARLHF. Additionally, the authors conduct various experiments to demonstrate the effectiveness of the proposed method, even when applied to offline datasets lacking uniform coverage.

**Weaknesses:**

1. The discussion section on related works is incomplete. The authors should provide a more thorough discussion of recent advancements in MARL and offline RLHF ([1]-[4]). Additionally, the paper emphasizes the importance of incorporating reward regularization in the objective function for the current task. However, similar ideas have been adopted in different contexts and should be discussed carefully ([3]-[6]).

2. The current experiments primarily showcase different variants of the proposed methods and include an ablation study. Could the authors include more baseline methods for comparison? Additionally, incorporating more tasks (e.g., five tasks) would strengthen the findings and provide greater convincing power for readers.

3. The theoretical analysis currently focuses solely on the linear function approximation setting, which may not be realistic given the use of neural networks in the experiments. Could the authors extend the analysis to accommodate general function approximations, or clarify how the experimental setup meets the requirements of linear function approximation?

4. In Line 300, it seems that someone even left comments colored in blue, which may leak the information of the authors. It is suggested that the authors should double-check the submitted draft to avoid this careless mistake.

5. In Line 276, the reference to "an approximate Nash equilibrium policy" in the theorem lacks clarity, as it does not illustrate the approximation error in relation to the size of the offline dataset. The authors should expand on the implications of the derived bound and compare their results with existing theoretical findings in the offline RL and MARL literature.


[1]  Wang, Yuanhao, et al. "Breaking the curse of multiagency: Provably efficient decentralized multi-agent rl with function approximation." The Thirty Sixth Annual Conference on Learning Theory. PMLR, 2023.

[2] Xiong, Nuoya, et al. "Sample-Efficient Multi-Agent RL: An Optimization Perspective." The Twelfth International Conference on Learning Representations.

[3] Liu, Zhihan, et al. "Provably mitigating overoptimization in rlhf: Your sft loss is implicitly an adversarial regularizer." arXiv preprint arXiv:2405.16436 (2024).

[4] Cen, Shicong, et al. "Value-Incentivized Preference Optimization: A Unified Approach to Online and Offline RLHF." arXiv preprint arXiv:2405.19320 (2024).

[5] Mete, Akshay, et al. "Reward biased maximum likelihood estimation for reinforcement learning." Learning for Dynamics and Control. PMLR, 2021.

[6] Xie, Tengyang, et al. "Bellman-consistent pessimism for offline reinforcement learning." Advances in neural information processing systems 34 (2021): 6683-6694.

**Questions:**

1. This paper analyzes the RLHF setting; however, the definition of the performance metric remains unchanged from the RL setting without KL regularization. Could the authors provide further clarification on this?

2. Could the authors highlight the novel aspects of the current theoretical analysis that differentiate it from the offline MARL setting?

---

> ### Author Response · Authors · 2024-11-25
>
> #### W1. Reward regularization idea in previous works
> Our work introduces reward regularization specifically within the context of preference-based MARL, addressing challenges due to the complexity of multi-agent interactions and long-horizon dependencies. While different from prior works that use adversarial adjustments [5] or Bellman-consistent pessimism [6] in general RL settings, we use MSE loss for reward regularization. However, we share a similar high-level idea to stabilize reward model training. We have incorporated these insights into the related work section.
>
> #### W2. Could the authors include more baseline methods and more tasks for comparison?
> As an initial work in preference-based MARL, to the best of our knowledge, there are **no directly comparable traditional approaches**.
> The most straightforward comparison is to combine a reward model with offline RL algorithms. In the updated paper, experiments on Overcooked are added. For other algorithms, experiments with MAIQL and MABCQ are also included. These two algorithms are commonly used in offline MARL papers. See results below and more details in Section 6 of the updated paper.
>
> #### W3. Could the authors extend the analysis to accommodate general function approximations, or clarify how the experimental setup meets the requirements of linear function approximation?
> We agree that our theory based on linear assumptions does not perfectly match our experiments. However, we want to emphasize that our theory provides a fundamental understanding of offline preference-based MARL and guided the experiment design, where we compared datasets with different levels of diversity. Extending our analysis to general function approximation would be an interesting future direction.
>
> #### W4. In Line 276, the reference to "an approximate Nash equilibrium policy" in the theorem lacks clarity
> We provided the complete theorem in the appendix (Theorem 4 in Line 1074). The approximation error is bounded by the inverse of the covariance norm, which will have an $O(\sqrt{n})$ rate if we have $n$ samples from a fixed distribution. This type of bound is also widely adopted in offline RL literature [1, 2].
>
> #### W5 & Q2. The authors should expand on the implications of the derived bound and compare their results with existing theoretical findings in the offline RL and MARL literature.
> For preference-based MARL, we extend the analysis to accommodate preference-based datasets, which differ from standard offline datasets with fixed state-action-reward tuples. This leads to different covariance matrices and uncertainty measures compared with standard offline MARL. Additionally, we establish that unilateral coverage is sufficient for learning approximate Nash equilibria, deriving bounds that explicitly account for preference-based dynamics.
>
> #### Q1. This paper analyzes the RLHF setting; however, the definition of the performance metric remains unchanged from the RL setting without KL regularization. Could the authors provide further clarification on this?
> In our study, we adopt the standard performance metric from traditional RL without incorporating KL regularization. We aim to propose a basic setting for preference-based MARL, aligning with the general cases. As KL regularization is an efficient practical method, it is also natural in RL literature to use a performance metric without additional regularization terms [3, 4]. We acknowledge that incorporating KL regularization can be beneficial in certain contexts, and we plan to explore its integration in future work.
>
> #### Typos and writing clarity
> Thank you for pointing out the typos and writing clarity problems. We have removed the blue words.
> We have enriched our related work section with the recent advancements in MARL and RLHF as you mentioned.
>
> > [1] Jin, Ying, Zhuoran Yang, and Zhaoran Wang. "Is pessimism provably efficient for offline rl?." International Conference on Machine Learning. PMLR, 2021.
> > [2] Zhong, Han, et al. "Pessimistic minimax value iteration: Provably efficient equilibrium learning from offline datasets." International Conference on Machine Learning. PMLR, 2022.
> > [3] Longyang Huang, Botao Dong, Ning Pang, et al. Offline Reinforcement Learning without Regularization and Pessimism. TechRxiv. June 07, 2024.
> > [4] Le Lan, Charline, Marc G. Bellemare, and Pablo Samuel Castro. "Metrics and continuity in reinforcement learning." Proceedings of the AAAI Conference on Artificial Intelligence. Vol. 35. No. 9. 2021.
> > [5] Mete, Akshay, et al. "Reward biased maximum likelihood estimation for reinforcement learning." Learning for Dynamics and Control. PMLR, 2021.
> > [6] Xie, Tengyang, et al. "Bellman-consistent pessimism for offline reinforcement learning." Advances in Neural Information Processing Systems 34 (2021): 6683-6694.

---

> > ### Author Response · Authors · 2024-11-25
> >
> > details in Section 6 of the updated papaer.
> >
> > | Algorithm                | Dataset          | Spread-v3        | Reference-v3     | Overcooked        |
> > |--------------------------|------------------|------------------|------------------|-------------------|
> > | **VDN with Pessimism Penalty** | Diversified      | -21.16 ± 0.54    | -18.89 ± 0.60    | **238.89 ± 3.50** |
> > |                          | Mix-Unilateral   | -21.03 ± 0.44    | -18.80 ± 0.63    | 221.80 ± 26.66    |
> > |                          | Mix-Expert       | -20.98 ± 0.54    | -18.80 ± 0.44    | 35.26 ± 55.19     |
> > |                          | Pure-Expert      | -21.01 ± 0.57    | -28.97 ± 2.89    | 3.36 ± 7.19       |
> > | **MAIQL**                | Diversified      | -25.33 ± 1.40    | -22.15 ± 0.55    | **16.59 ± 11.22** |
> > |                          | Mix-Unilateral   | -23.25 ± 1.06    | -23.22 ± 1.37    | 0.00 ± 0.00       |
> > |                          | Mix-Expert       | -23.26 ± 0.90    | -24.21 ± 1.60    | 0.00 ± 0.00       |
> > |                          | Pure-Expert      | -26.01 ± 1.53    | -29.47 ± 1.65    | 0.00 ± 0.00       |
> > | **MABCQ**                | Diversified      | -20.02 ± 0.64    | -17.64 ± 0.43    | **239.34 ± 1.67** |
> > |                          | Mix-Unilateral   | -19.47 ± 0.33    | -17.64 ± 1.11    | 215.01 ± 65.43    |
> > |                          | Mix-Expert       | -19.42 ± 0.17    | -17.88 ± 0.78    | 50.32 ± 82.82     |
> > |                          | Pure-Expert      | -20.56 ± 0.38    | -25.90 ± 1.11    | 1.14 ± 3.46       |

---

> > > ### Comment · Reviewer_oCgR · 2024-12-03
> > > **Reply to the Authors**
> > >
> > > Thank the authors for their replies and my concerns are partially solved. I am still not fully convinced why the current paper does not study the general function approximation setting and compare with other multiagent algorithms such as [1]. Hence, I would keep my score.
> > >
> > > [1] Yu, C., et al. "The surprising effectiveness of ppo in cooperative, multi-agent games. arXiv 2021." arXiv preprint arXiv:2103.01955.

---

> > > > ### Author Response · Authors · 2024-12-03
> > > >
> > > > Thank you for your feedback!
> > > > #### General function approximation
> > > > We believe that extending our result to general function follows the standard treatment of replacing linear function approximation with general function approximation. For example, it is straightforward to adapt the analysis for general function approximation in [1] to our framework.
> > > > #### Comparison with other multi-agent algorithms like MAPPO
> > > > We did not test on-policy algorithms like PPO because they require the replay buffer to be collected by the currently optimized policy, making them unsuitable for direct application in offline reinforcement learning.
> > > >
> > > > [1] Zhang et al. "Offline Learning in Markov Games with General Function Approximation." https://arxiv.org/pdf/2302.02571

---

### Official Review · Reviewer_Zpkc · 2024-11-03

**Soundness:** 3
**Presentation:** 2
**Contribution:** 2
**Rating:** 5
**Confidence:** 4

**Summary:**

The paper addresses the problem of trying to learn human preferences (this behaviour is better than that behaviour) in a multi agent RL setup. In this case satisfactory learning means a Nash-equilibrium is reached between all policies. The authors positions the paper as an initial study into Multiagent Reinforcement Learning from Human Feedback.

The paper shows how pure expert policies are not always the best for maximising overall score, and that mixing in less expert policies in some cases causes an overall higher score to be reached in the MARLHF case. This is proved, theoretically. They also show that it is often easier to learn what policies score higher by having unilaterally divergent policies acting in the environment, where a single agent is using a sub-optimal policy. The authors call this approach unilateral coverage. By having this unilateral agent in the environment it becomes simpler to observe what policies may be truly optimal within the environment. In addition upper complexity bounds are established for Nash Equilibrium in effective MARLHF.

The process to implement this approach is to learn a reward function from a preference dataset while mitigating extrapolation errors with a pessimism term and then determining a final policy. Human Feedback is itself simulated using the Bradley-Terry-Luce model to rank solutions.

The authors make 2 particular contributions to implement their insights:
Applying MSE regularisation to the training data to distribute rewards more evenly across timesteps, which helps to avoid temporal concentration. This essentially takes the sparse reward signals from the Bradley-Terry-Luce model and spread them out to produce reward over more timesteps.
Dataset distribution-based penalties are used to constrain exploration to known regions of the state space

Their empirical evaluation spans three multi-agent scenarios: cooperative target coverage, coordinated pursuit, and communication-dependent navigation. They show that incorporating imperfect policies is helpful for learning higher scoring policies during training. In harder tasks, unilateral coverage and diversity become more important and more diverse datasets led to lower variance in training outcomes. The authors also introduce a principled standardization technique for hyperparameter tuning across environments.

**Strengths:**

In terms of the proofs, there is a simple but convincing proof by counterexample provided for theorem 1 (not contradiction, as stated).
There is an explicit bounds found on the Nash-gap.

Hyperparameters used in the training are provided, multiple seeds are used and results that don’t support the desired conclusion are presented. Multiple environments are tested, and clear ablation studies are done.

The paper makes an interesting theoretical contribution by establishing fundamental results about Multi-Agent Reinforcement Learning from Human Feedback (MARLHF). The authors prove why single-policy coverage is insufficient and demonstrate that unilateral coverage is both necessary and sufficient for learning Nash equilibria. These theoretical foundations are presented with clear proofs that are well constructed. These theoretical results then explicitly inform the design of the framework which is clearly stated and explained.

The empirical work is comprehensive and well-designed, testing their approach across three distinct multi-agent scenarios that each present different challenges (cooperative target coverage, coordinated pursuit, and communication-dependent navigation). The experiments validate both the theoretical insights about dataset coverage and the effectiveness of their algorithmic innovations. Their ablation studies are thorough and give clear evidence for the value of their MSE regularization and dataset distribution-related penalties. The authors also introduce a practical standardization technique for hyperparameter tuning that works across different environments.

The clarity of the experimental setup makes the work also highly reproducible

**Weaknesses:**

The main weakness is that despite the paper's title and framing, there is no actual human feedback involved in any of the experiments. Instead, the authors simulate preferences using the Bradley-Terry-Luce model based on known reward functions from the environments. This is a significant limitation because real human preferences are likely to be much noisier, inconsistent, and potentially non-transitive compared to their simulated preferences. The paper would be more accurately titled as "Multi-Agent Reinforcement Learning from Simulated Preferences" or similar, and should more explicitly acknowledge this limitation and discuss how their approach might need to be modified for real human feedback.

While thorough, the theoretical results rely heavily on assumptions that may not hold in practice. The paper assumes linear Markov games and works with known feature mappings, but doesn't discuss enough how these assumptions might limit real-world applicability. Additionally, although the paper proves that their theoretical algorithm converges to Nash equilibria, the practical implementation uses different algorithms (VDN-based) with no theoretical guarantees. This gap between theory and practice is not sufficiently discussed. The paper also doesn't explore whether the Nash equilibrium is actually desirable in all cases - in some scenarios, other solution concepts might better align with human preferences. This again is one of the major weaknesses with the unclear framing.

The experimental evaluation, while systematic, is limited to relatively simple environments in the Multi-Agent Particle Environment (MPE) framework. These environments, while useful for testing basic concepts, are far simpler than real-world multi-agent scenarios. The paper doesn't adequately discuss how their approach might scale to more complex environments or to scenarios with larger numbers of agents. Their results showing that mixed-skill policies can outperform pure expert policies raise questions about whether their reward modeling approach is capturing the true objectives of the tasks.

Another important weakness in the paper's empirical evaluation is the absence of statistical significance testing. Although results with means and standard deviations across 5 random seeds are given, they don't perform any statistical analysis to validate the conclusions. This is particularly problematic given the small sample size - with only 5 seeds, the reliability of their comparisons is questionable. The paper lacks hypothesis tests. This makes it difficult to determine if the reported differences between approaches are statistically significant, especially in cases where the differences appear small relative to their standard deviations. For example, in Spread-v3, it's unclear whether the difference between "Mix-Unilateral" (-20.98 ± 0.56) and "Mix-Expert" (-21.11 ± 1.16) is meaningful. The lack of statistical rigor undermines the strength of the paper's empirical conclusions and the claims made about the benefits of their approaches.

**Questions:**

How would your approach need to be modified to handle inconsistent or non-transitive preferences that often occur with real human feedback?
Why do you call the paper MARLHF when there is clearly no HF?
The practical implementation differs significantly from the theoretical algorithm - can you explain this gap and discuss whether any theoretical guarantees carry over?
Given the relative simplicity of the tasks, why were only 5 random seeds used for the experiments?
Why weren't statistical significance tests performed to validate the comparative results?
How well does your approach scale with increasing numbers of agents?
In cases where mixed-skill policies outperform pure expert policies, can you verify that this reflects genuine improvement rather than issues with reward modeling?
Have you tested MARL algorithms other than VDN?

---

> ### Author Response · Authors · 2024-11-25
>
> Thank you for your review. Please find our responses to your comments below.
>
> #### WP1. Q1. & Q2. There is no actual human feedback involved.
> You are correct that our current experiments rely on simulated preferences rather than true human feedback. To better reflect this, we will change the title to "Preference-Based Multi-Agent Reinforcement Learning" to clarify our focus.
>
> #### WP2. The theoretical results rely heavily on assumptions that may not hold in practice.
> We agree that our theory based on linear assumptions does not perfectly match our experiments. However, we want to emphasize that our theory provides a fundamental understanding of offline preference-based MARL and guided the experiment design, where we compared datasets with different levels of diversity. Extending our analysis to general function approximation would be an interesting future direction.
>
> #### Q3. The paper proves that their theoretical algorithm converges to Nash equilibria, but the practical implementation uses different algorithms (VDN-based) with no theoretical guarantees.
> Our theory demonstrates the data coverage conditions fundamentally needed even in the basic linear function approximation setting. For more real-world problems requiring general function approximation, such as neural networks, we chose VDN to solve these problems. Our experiments showed that unilateral coverage and more diversified datasets improve performance, verifying our theoretical insights.
>
> #### WP4. & Q4. With only 5 seeds, the reliability of their comparisons is questionable.
> Thank you for raising this point. We have rerun our experiments with 10 seeds and updated the data in the revised version. We observed that the variance in performance across different seeds remains very low, which we believe provides sufficient robustness for our findings.
>
> #### WP4. & Q5. The differences appear small relative to their standard deviations.
> Some of the comparative results indeed lack persuasiveness, such as the examples you mentioned (-20.98 ± 0.56) and (-21.11 ± 1.16). As a result, our practice of bolding data with only minor advantages in the table may have been misleading. In the revised version, only comparison results with a p-value less than 0.05 in the significance test are bolded. Additionally, more precise language is used to describe the empirical conclusions.
>
> #### Q6. How well does your approach scale with increasing numbers of agents?
> We've conducted an experiment to test the scaling problem in Appendix B3 (B4 in the updated version). A brief conclusion is that, while our current approach manages the scaling of agents without introducing new problems, it does not specifically address the inherent issues of instability and complexity that are well-documented in traditional MARL. We added a paragraph discussing this in the main text of the updated paper.
>
> #### Q7. In cases where mixed-skill policies outperform pure expert policies, can you verify that this reflects genuine improvement rather than issues with reward modeling?
> In preference-based RL, reward modeling is an integral part of the complete algorithm, and our theoretical analysis does not separate reward modeling from the subsequent RL Oracle. For example, in Theorem 4 (line 1074), the approximation error is bounded by the inverse of the covariance norm, which depends on the diversity of the dataset. Therefore, the fact that a mixed dataset can provide a better reward model than a pure dataset is itself a genuine improvement.
>
> #### Q8. & WP3. Have you tested MARL algorithms other than VDN? & The experimental evaluation, while systematic, is limited to relatively simple environments in the Multi-Agent Particle Environment (MPE) framework.
> Thank you for the advice! In the updated paper, experiments on Overcooked are added. For other algorithms, experiments with MAIQL and MABCQ are added. These two algorithms are commonly used in offline MARL papers. See results below and more details in Section 6 of the updated paper.

---

> > ### Author Response · Authors · 2024-11-25
> >
> > | Algorithm                | Dataset          | Spread-v3        | Reference-v3     | Overcooked        |
> > |--------------------------|------------------|------------------|------------------|-------------------|
> > | **VDN with Pessimism Penalty** | Diversified      | -21.16 ± 0.54    | -18.89 ± 0.60    | **238.89 ± 3.50** |
> > |                          | Mix-Unilateral   | -21.03 ± 0.44    | -18.80 ± 0.63    | 221.80 ± 26.66    |
> > |                          | Mix-Expert       | -20.98 ± 0.54    | -18.80 ± 0.44    | 35.26 ± 55.19     |
> > |                          | Pure-Expert      | -21.01 ± 0.57    | -28.97 ± 2.89    | 3.36 ± 7.19       |
> > | **MAIQL**                | Diversified      | -25.33 ± 1.40    | -22.15 ± 0.55    | **16.59 ± 11.22** |
> > |                          | Mix-Unilateral   | -23.25 ± 1.06    | -23.22 ± 1.37    | 0.00 ± 0.00       |
> > |                          | Mix-Expert       | -23.26 ± 0.90    | -24.21 ± 1.60    | 0.00 ± 0.00       |
> > |                          | Pure-Expert      | -26.01 ± 1.53    | -29.47 ± 1.65    | 0.00 ± 0.00       |
> > | **MABCQ**                | Diversified      | -20.02 ± 0.64    | -17.64 ± 0.43    | **239.34 ± 1.67** |
> > |                          | Mix-Unilateral   | -19.47 ± 0.33    | -17.64 ± 1.11    | 215.01 ± 65.43    |
> > |                          | Mix-Expert       | -19.42 ± 0.17    | -17.88 ± 0.78    | 50.32 ± 82.82     |
> > |                          | Pure-Expert      | -20.56 ± 0.38    | -25.90 ± 1.11    | 1.14 ± 3.46       |

---

### Official Review · Reviewer_1Zjc · 2024-11-03

**Soundness:** 2
**Presentation:** 3
**Contribution:** 3
**Rating:** 6
**Confidence:** 4

**Summary:**

This paper investigates the important and timely problem of multi-agent reinforcement learning from human feedback (MARLHF). The authors examine both theoretical and practical aspects of MARLHF, demonstrating that single policy coverage is insufficient and emphasizing the need for unilateral dataset coverage. To address the issues of sparse and spiky reward learning typical in standard RLHF, they propose two primary techniques: (1) mean squared error regularization to promote uniform reward distribution, and (2) an additional reward term based on state-action pair density within the dataset to introduce pessimism, using an imitation learning-based approach for density modeling. The final policy is then trained using the VDN algorithm. Overall, this MARLHF approach represents a significant step toward preference-based reinforcement learning in multi-agent systems.

**Strengths:**

* This paper makes novel contributions to RLHF within multi-agent systems by framing the task as finding a Nash equilibrium in general-sum games and introducing innovative techniques for reward regularization and dataset distribution-based pessimism.
* The theoretical results are comprehensive and well-justified, effectively supporting the paper’s claims.
* The paper is generally well-written and easy to follow.

**Weaknesses:**

* The empirical validation of the approach is limited, as the paper only includes experiments on three simple MPE environments. Since the authors utilized JAXMARL, testing on more realistic and complex environments from the JAXMARL API, such as Overcooked, Hanabi, or StarCraft, would strengthen the paper’s claims.
* The comparison with MARL baselines is insufficient, focusing only on VDN despite its known limitations in representation capacity. Conducting ablation studies with other MARL algorithms, such as MAPPO[1], IPPO[2], and QMIX[3], would provide more validations.

**Questions:**

1. Why was VDN specifically chosen as the base MARL algorithm, given its known limitations in representation capacity? How would the proposed approach perform with more advanced MARL algorithms like MAPPO, IPPO, or QMIX?
2. Given that the experiments were conducted only on MPE environments (Spread-v3, Tag-v3, Reference-v3), how would the method perform on more complex MARL benchmarks? What challenges do you anticipate, and how sensitive might performance be to the choice of hyperparameters $\alpha$ and $\beta$?
3. What policy was used to generate responses for collecting preference feedback?
4. How was the preference feedback collected? Was it synthetic, based on true environment rewards, or did it come from real human preferences? These details are crucial for reproducibility, a deeper understanding of the approach, and identifying potential biases in the preference data.
5. The inherent dependence between the policy used to train the reward model and the policy being learned is not addressed in the paper. For instance, in the single-agent setting (see [4]), this dependence can be significant. How does the proposed approach handle this issue?
6. How does the quality of the learned reward function vary with different levels of expertise and sparsity in preference feedback?

[1] Yu, Chao, et al. "The surprising effectiveness of ppo in cooperative multi-agent games." Advances in Neural Information Processing Systems 35 (2022): 24611-24624.

[2] De Witt, Christian Schroeder, et al. "Is independent learning all you need in the starcraft multi-agent challenge?." arXiv preprint arXiv:2011.09533 (2020).

[3] Rashid, Tabish, et al. "Monotonic value function factorisation for deep multi-agent reinforcement learning." Journal of Machine Learning Research 21.178 (2020): 1-51.

[4]  Chakraborty, Souradip, et al. "PARL: A Unified Framework for Policy Alignment in Reinforcement Learning from Human Feedback." The Twelfth International Conference on Learning Representations.

---

> ### Author Response · Authors · 2024-11-25
>
> Thank you for your review. Please find our responses to all your comments below.
>
> #### W1. W2. & Q1. Testing on more realistic and complex environments and conducting ablation studies with other MARL algorithms
>
> Thank you for the advice! In the updated paper, we added the Overcooked environment to address the lack of long-duration, strategy-intensive task environments. Furthermore, we additionally tested two more algorithms: Multi-agent IQL (MAIQL) [1] and Multi-agent BCQ (MABCQ) [2].
>
> - Remark 1: The results on Overcooked highlight the importance of dataset diversity and unilateral data.
>
> - Remark 2: Since preference-based MARL is already a new and complex setup, we intentionally selected simple implementations for both the environment and the algorithms to avoid introducing additional uncertainties. We did not test on-policy algorithms like PPO because they require the replay buffer to be collected by the currently optimized policy, making them unsuitable for direct application in offline reinforcement learning, which is the focus of our setting. On the other hand, the QMIX algorithm is more suited for environments involving a large number of agents and complex policies.
> Therefore, for additional experiments with more algorithms, we adopted Multi-agent IQL (MAIQL) [1] and Multi-agent BCQ (MABCQ) [2], which are simpler to implement but commonly used in offline MARL papers under the centralized-training decentralized-execution (CTDE) framework.
>
> #### Q2. What challenges do you anticipate on more complex MARL benchmarks?
> One of the primary challenges is reward modeling. In more complex environments (e.g., SMAC), especially in longer-horizon, continuous settings, the lack of supervisory signals becomes more apparent. Since reward models learned through preference learning often exhibit significant errors, agents tend to perform poorly in low-robustness tasks that emphasize fine-grained operations. This contrast is evident in the experimental results of MPE and the newly added Overcooked environment: in Overcooked, the agents can generally achieve performance close to the expert level, whereas in the three MPE environments, there is a noticeable gap between the two.
>
> #### Q2. How sensitive might performance be to the choice of hyperparameters $\alpha$ and $\beta$?
> An ablation study on $\beta$ and $\alpha$ can be found in Section 6 and Appendix B.5.
> The choice of $\beta$ is highly robust. Due to clipping, excessively large $\beta$ values will not dominate the entire reward function. As a result, larger $\beta$ values almost never degrade the agent's performance in our experiments. This allows us to increase $\beta$ with relative confidence. Therefore, we generally recommend setting $\beta$ to a value between 10 and 100.
> | $\beta$             | 0               | 0.1             | 1               | 10              | 100             |
> |---------------|-----------------|-----------------|-----------------|-----------------|-----------------|
> | Spread-v3     | -22.56 ± 1.61   | -22.03 ± 0.67   | -20.82 ± 0.53   | -20.46 ± 0.51   | -20.35 ± 0.43   |
> | Tag-v3        | 4.11 ± 1.66     | 4.25 ± 0.53     | 10.96 ± 1.20    | 28.88 ± 1.02    | 29.53 ± 1.35    |
> | Reference-v3  | -19.69 ± 0.36   | -19.37 ± 0.53   | -18.89 ± 0.78   | -18.33 ± 0.42   | -18.54 ± 0.46   |
> | Overcooked    | 0.00 ± 0.00     | 0.00 ± 0.00     | 149.53 ± 86.74  | 238.89 ± 3.50   | **240 ± 0.00**  |
> The choice of $\alpha$, however, is more nuanced. Lower $\alpha$ values tend to reduce the reward model's loss, but since smoother curves are often more suitable for learning, higher reward model losses can sometimes lead to better RL training results. Empirically, setting $\alpha$ to 1 gives near-optimal results.
>
> | $\alpha$       | 0     | 0.001 | 0.01  | 0.1   | 1     | 10    | 100   | 1000  |
> |----------------|-------|-------|-------|-------|-------|-------|-------|-------|
> | Spread-v3      | 0.350 | 0.345 | 0.347 | 0.351 | 0.361 | 0.389 | 0.460 | 0.603 |
> | Tag-v3         | 0.465 | 0.431 | 0.440 | 0.455 | 0.484 | 0.531 | 0.603 | 0.676 |
> | Reference-v3   | 0.358 | 0.356 | 0.362 | 0.374 | 0.393 | 0.434 | 0.508 | 0.623 |
>
> #### Q3. What policy was used to generate responses for collecting preference feedback?
> We used VDN for the three MPE environments and IPPO on Overcooked to train the expert policy. We tested VDN, IPPO, MAPPO, and QMIX on all the environments and took the best agent trained as the expert. Suboptimal trajectories were collected using checkpoints saved during the training of the expert policy.
> More details can be found in Section 6.
>
> #### Q4. How was the preference feedback collected?
> We simulated the preferences with the Bradley-Terry Model. Details can be found in Section 6.

---

> > ### Author Response · Authors · 2024-11-25
> >
> > #### Q5. The inherent dependence between the policy used to train the reward model and the policy being learned.
> >
> > The PARL paper you mentioned and the problem setting in this paper are not entirely the same. In the PARL paper, the agent performs online learning at the end, whereas in our paper, learning is restricted to offline training with a fixed dataset.
> >
> > In offline reinforcement learning, since there are no performance guarantees outside the dataset, the learned policy is inherently constrained within the dataset, meaning it is "inherently dependent on the offline dataset." In other words, because both the reward model training and the agent training use the same dataset, we can expect that for states where the agent achieves a low Bellman error, the reward can also be well-estimated. Therefore, this issue does not exist in offline preference-based MARL.
> >
> > #### Q6. How does the quality of the learned reward function vary with different levels of expertise and sparsity in preference feedback?
> >
> > In our experiments, under the condition of maintaining the same level of diversity, higher-quality expert data resulted in better reward models. This is because the "positive features" in demonstrations from better experts are more prominent, with fewer irrelevant noise or erroneous demonstrations. Specifically, using the return difference between the final trained agent and the expert as a standard, suboptimal experts lead to significantly larger discrepancies compared to the best experts. Therefore, we primarily used the most expert-level demonstrations in our main experiments.
> >
> > Since we only have preferences between complete trajectories, the sparsity of preference feedback is fixed at 1 preference per episode pair.
> >
> > | Algorithm                | Dataset          | Spread-v3        | Reference-v3     | Overcooked        |
> > |--------------------------|------------------|------------------|------------------|-------------------|
> > | **VDN with Pessimism Penalty** | Diversified      | -21.16 ± 0.54    | -18.89 ± 0.60    | **238.89 ± 3.50** |
> > |                          | Mix-Unilateral   | -21.03 ± 0.44    | -18.80 ± 0.63    | 221.80 ± 26.66    |
> > |                          | Mix-Expert       | -20.98 ± 0.54    | -18.80 ± 0.44    | 35.26 ± 55.19     |
> > |                          | Pure-Expert      | -21.01 ± 0.57    | -28.97 ± 2.89    | 3.36 ± 7.19       |
> > | **MAIQL**                | Diversified      | -25.33 ± 1.40    | -22.15 ± 0.55    | **16.59 ± 11.22** |
> > |                          | Mix-Unilateral   | -23.25 ± 1.06    | -23.22 ± 1.37    | 0.00 ± 0.00       |
> > |                          | Mix-Expert       | -23.26 ± 0.90    | -24.21 ± 1.60    | 0.00 ± 0.00       |
> > |                          | Pure-Expert      | -26.01 ± 1.53    | -29.47 ± 1.65    | 0.00 ± 0.00       |
> > | **MABCQ**                | Diversified      | -20.02 ± 0.64    | -17.64 ± 0.43    | **239.34 ± 1.67** |
> > |                          | Mix-Unilateral   | -19.47 ± 0.33    | -17.64 ± 1.11    | 215.01 ± 65.43    |
> > |                          | Mix-Expert       | -19.42 ± 0.17    | -17.88 ± 0.78    | 50.32 ± 82.82     |
> > |                          | Pure-Expert      | -20.56 ± 0.38    | -25.90 ± 1.11    | 1.14 ± 3.46       |
> >
> >
> > > [1] Kostrikov et al., Offline reinforcement learning with implicit Q-learning. https://arxiv.org/abs/2110.06169
> > > [2] Fujimoto et al., Off-policy deep reinforcement learning without exploration. https://arxiv.org/abs/1812.02900

---

> > > ### Comment · Reviewer_1Zjc · 2024-11-26
> > >
> > > Thank you for your thorough and detailed response. Authors have addressed most of my concerns, and I am happy to increase my scores to reflect the improvements made.

---

> > > > ### Author Response · Authors · 2024-12-04
> > > >
> > > > Thank you for your feedback! We greatly appreciate your time and effort in reviewing our paper and are encouraged by your updated evaluation.

---

### Author Response · Authors · 2024-11-25
**General Response**

### **Revisions in the Updated Paper**

Revisions are marked in blue in the updated paper:
- **"MARLHF"** is replaced with **"PbMARL" (Preference-based MARL)**.
- The **Related Work** section is enriched.
- Missing details about **dataset distribution-based pessimism** are added.
- All the experiments in the original paper are **rerun** with 10 seeds.
- Experiments in the **Overcooked** environment and corresponding descriptions are added.
- Experiments with **MABCQ** and **MACQL** and their corresponding descriptions are added.
- Comments on **empirical results** are updated based on the new experiments.
- A paragraph about scalability is added to the **Experiments** section.
- **Typos are fixed**.

---

### **Acknowledgment of Reviewers' Feedback**

**We thank all the reviewers for their insightful and constructive feedback!**
We are encouraged by their positive evaluation of our work and their recognition of our contributions. We appreciate the reviewers' acknowledgment of our strong theoretical framework, including novel contributions to modeling MARLHF and establishing a solid theoretical foundation (1Zjc, Zpkc, oCgR, TKUr).

We are particularly pleased that the reviewers found our empirical validation comprehensive and well-designed, with experiments that align with and verify our theoretical claims (Zpkc, oCgR, TKUr). The effective empirical techniques we proposed, such as reward regularization and dataset distribution-based pessimism, were also noted for their impact on stabilizing learning and improving performance (1Zjc, oCgR).

We have carefully considered all the feedback and suggestions provided, addressing specific reviewer comments below, and will incorporate their valuable suggestions into our paper.

---

### **Response to a Common Concern: Preference Feedback**

As multiple reviewers (1Zjc, Zpkc) have asked about details regarding preference feedback, particularly the lack of human feedback, we address this inquiry here.

#### **Title Change**
First of all, to clarify our focus, we have changed our title to **"Preference-Based Multi-Agent Reinforcement Learning: Data Coverage and Algorithmic Techniques"**.

#### **Using a Gold Reward Model**
Using a gold reward model when constructing the dataset is a standard practice in the RLHF literature. An observation in [1] shows that first learning a proxy reward and then passing it into the BT model for pseudo-labeling to construct a new dataset outperforms directly learning from offline data. Section 6.1 of [2] describes an environment where offline data is directly constructed using a reward function. Similarly, [3] conducts experiments on datasets constructed from a reward model.

As a first step to advancing our understanding of MARLHF, our work primarily focuses on **bridging theories with experiments**, given limited computational resources. We agree that it is a valuable direction to construct datasets from more complex tasks, such as language modeling using preferences from human or AI feedback, and we leave this for future work.

#### **Modeling Non-Transitive Preferences**
One of our key techniques, **reward regularization**, is specifically designed for a reward-based preference setting. Without this technique, it is challenging to handle trajectory data in Markov games due to long horizons, making accurate credit assignment infeasible. Even in RLHF for Markov decision processes, it is difficult to address non-transitive preferences. For example, [4] models non-transitive preferences using a min-max game. Extending this to Markov games is non-trivial and requires careful design.

---

### **New Experiments**

As all reviewers mentioned the limited number of environments tested, and multiple reviewers (1Zjc, oCgR) requested comparisons with more algorithms, we have significantly updated the experiment section with new environments and algorithms.

We reran all experiments with **10 different seeds**, added a new environment (**Overcooked**), and introduced two additional algorithms (MAIQL and MABCQ). The main results are summarized in the table below. MAIQL and MABCQ are the CTDE versions of IQL [5] and BCQ [6], respectively. The results strongly support our claims regarding the importance of **data diversity** and **unilateral data**.

For more detailed discussions and results, please refer to **Table 2** and **Table 3** in the updated paper.
> [1] Xiong et. al., Iterative preference learning from human feedback: Bridging theory and practice for RLHF under KL-constraint.
> [2] Song et. al., The Importance of Online Data: Understanding Preference Fine-tuning via Coverage.
> [3] Tajwar et. al., Preference Fine-Tuning of LLMs Should Leverage Suboptimal, On-Policy Data.
> [4] Swamy et. al., A Minimaximalist Approach to Reinforcement Learning from Human Feedback.
> [5] Kostrikov et. al., Offline reinforcement learning with implicit q-learning.
> [6] Fujimoto et. al. Off-policy deep reinforcement learning without exploration.

---

> ### Author Response · Authors · 2024-11-25
> **New experiment results**
>
> | Algorithm                | Dataset          | Spread-v3        | Reference-v3     | Overcooked        |
> |--------------------------|------------------|------------------|------------------|-------------------|
> | **VDN with Pessimism Penalty** | Diversified      | -21.16 ± 0.54    | -18.89 ± 0.60    | **238.89 ± 3.50** |
> |                          | Mix-Unilateral   | -21.03 ± 0.44    | -18.80 ± 0.63    | 221.80 ± 26.66    |
> |                          | Mix-Expert       | -20.98 ± 0.54    | -18.80 ± 0.44    | 35.26 ± 55.19     |
> |                          | Pure-Expert      | -21.01 ± 0.57    | -28.97 ± 2.89    | 3.36 ± 7.19       |
> | **MAIQL**                | Diversified      | -25.33 ± 1.40    | -22.15 ± 0.55    | **16.59 ± 11.22** |
> |                          | Mix-Unilateral   | -23.25 ± 1.06    | -23.22 ± 1.37    | 0.00 ± 0.00       |
> |                          | Mix-Expert       | -23.26 ± 0.90    | -24.21 ± 1.60    | 0.00 ± 0.00       |
> |                          | Pure-Expert      | -26.01 ± 1.53    | -29.47 ± 1.65    | 0.00 ± 0.00       |
> | **MABCQ**                | Diversified      | -20.02 ± 0.64    | -17.64 ± 0.43    | **239.34 ± 1.67** |
> |                          | Mix-Unilateral   | -19.47 ± 0.33    | -17.64 ± 1.11    | 215.01 ± 65.43    |
> |                          | Mix-Expert       | -19.42 ± 0.17    | -17.88 ± 0.78    | 50.32 ± 82.82     |
> |                          | Pure-Expert      | -20.56 ± 0.38    | -25.90 ± 1.11    | 1.14 ± 3.46       |

---

> > ### Comment · Reviewer_Zpkc · 2024-11-25
> > **Response to rebuttal**
> >
> > Thank you to the authors. I believe that some improvements have been made and I am raising my score accordingly.

---

### Meta-Review · Area_Chair_uk29 · 2024-12-11

**Metareview:**

This paper investigates offline multi-agent RL with preference data. Theoretical study establishes the insufficiency of the single policy coverage and demonstrates the need for unilateral dataset coverage. From the empirical side, a MSE reward regularizer and a pessimistic penalty are proposed. While the reviewers acknowledge the theoretical contribution, I have the following concerns about the work.

1. Almost all the empirical results in the three small problems are not statistically significant. Essentially, this means that there is only one single environment in this paper -- overcooked. But as suggested by 1Zjc, JaxMARL does have other environments so I feel this paper at least needs a major revision to test on one more environment.

2. I failed to find any evidence that the MSE reward regularizer actually contributes positively to the final performance. In Table 4, it shows that the method has 240 scores in overcooked even with $\alpha=0$. In the reply to 1Zjc, there is an ablation study of $\alpha$ and $\beta$. But only $\beta$ is studied in overcooked. $\alpha$ is not. So I cannot find any evidence supporting the effectiveness of $\alpha$, the MSE reward regularizer. Figure 2(a) does not involve overcooked as well, which is the only env that this paper really should considers. The authors did not exclude the possibility that Figure 2(a) works because this regularizer overfits to the three small tasks.

3. The theory and the two regularizers (MSE reward regularizer and pessimistic penalty) are entirely disconnected. I cannot see sufficient  theoretical justification / motivation for those two regularizers. This is not a problem if the two work well. But again, this paper essentially has only 1 env and I cannot find evidence that the MSE reward regularizer actually matters in overcooked.

To summarize, I think the theoretical findings of this paper are interesting and promising. But the paper lacks sufficient empirical study to support the proposed regularizers. In fact, overcooked is only added during the rebuttal period. I feel the empirical part of the paper is not ready and needs at least a major revision. The paper may also benefit from connecting the two regularizers more with the proposed theory. I'm fine with doing theories with linear function approximation and doing experiments with neural networks. But the disconnection between theories and experiments in this submission now is far beyond linear function approximation and neural networks.

It is also worth mentioning that the authors accidentally removed the appendix entirely in the rebuttal period so I can only use the original appendix.

**Additional Comments On Reviewer Discussion:**

Before the AC-reviewer discussion period, this paper has 6666. I checked all the comments and read the paper myself and raised the three concerns during AC-reviewer discussion. Reviewers are convinced by my arguments and lowered their scores accordingly. And no reviewer is able / willing to argue for acceptance.

---

### Decision · Program_Chairs · 2025-01-22

Reject